# Bioefficacy of Nga-Mon (*Perilla frutescens*) Fresh and Dry Leaf: Assessment of Antioxidant, Antimutagenicity, and Anti-Inflammatory Potential

**DOI:** 10.3390/plants12112210

**Published:** 2023-06-03

**Authors:** Payungsak Tantipaiboonwong, Komsak Pintha, Wittaya Chaiwangyen, Maitree Suttajit, Chakkrit Khanaree, Orawan Khantamat

**Affiliations:** 1Division of Biochemistry, School of Medical Sciences, University of Phayao, Phayao 56000, Thailand; payungsak.t@gmail.com (P.T.); komsakjo@gmail.com (K.P.); wittaya.ch@up.ac.th (W.C.); maitree.suttajit@gmail.com (M.S.); 2School of Traditional and Alternative Medicine, Chiang Rai Rajabhat University, Chiang Rai 57100, Thailand; chakkrit.kh@gmail.com; 3Department of Biochemistry, Faculty of Medicine, Chiang Mai University, Chiang Mai 50200, Thailand

**Keywords:** *Perilla frutescens*, Nga-mon, antioxidant, antimutagenicity, anti-inflammation

## Abstract

Perilla leaves are known to be a rich source of polyphenols, which have been shown to exhibit various biological effects. This study aimed to compare the bioefficacies and bioactivities of fresh (PLE_f_) and dry (PLE_d_) Thai perilla (Nga-mon) leaf extracts. Phytochemical analysis indicated that both PLE_f_ and PLE_d_ were abundant in rosmarinic acid and bioactive phenolic compounds. PLE_d_, which had higher levels of rosmarinic acid but lower levels of ferulic acid and luteolin than PLE_f_, exhibited greater effectiveness in a free radical scavenging assay. Furthermore, both extracts were found to suppress intracellular ROS generation and exhibit antimutagenic activity against food-borne carcinogens in *S. typhimurium*. They also attenuated lipopolysaccharide-induced inflammation in RAW 264.7 cells by inhibiting the expression of nitric oxide, iNOS, COX-2, TNF-α, IL-1β, and IL-6 through the suppression of NF-κB activation and translocation. However, PLE_f_ exhibited a higher ability to suppress cellular ROS production and higher antimutagenic and anti-inflammatory activities than PLE_d_, which can be attributed to its combination of phytochemical components. Overall, PLE_f_ and PLE_d_ have the potential to serve as natural bioactive antioxidant, antimutagenic, and anti-inflammatory agents to achieve potential health benefits.

## 1. Introduction

Free radicals can cause oxidative stress, cellular damage, and inflammation, which may lead to metabolic disorders including cancers [1,2,3]. The pathogenesis of inflammation involves complex interactions between immune cells, cytokines, and pro-inflammatory genes [4]. The production of pro-inflammatory cytokines such as tumor necrosis factor-α (TNF-α), interleukin-1β (IL-1β), and interleukin-6 (IL-6), along with inflammatory mediators such as nitric oxide (NO) and prostaglandin E2 (PGE2), plays a crucial role in the inflammatory response [5,6]. Therefore, managing oxidative stress and reducing the overproduction of pro-inflammatory cytokines and mediators could be a promising strategy for alleviating inflammatory-related disorders.

Thai perilla (*Perilla frutescens*), also referred to as Nga-mon, is an herbaceous plant traditionally grown in Northern Thailand. Perilla leaves and seeds have various beneficial properties and are used in culinary and medicinal applications [7,8,9]. The bioactive phytoconstituents present in perilla leaves include polyphenols, flavonoids, vitamins, and essential fatty acids [7,10]. The concentration of phytochemicals in perilla leaves may differ due to multiple factors such as plant form, growing circumstances, and extraction techniques [11,12,13]. Perilla dried leaf extract is abundant in rosmarinic acid, which exhibits antioxidant, anti-inflammatory, and other therapeutic properties [13,14,15,16,17,18].

Recently, the use of Thai perilla leaf extract has become increasingly popular in various industries, such as the food, beverage, pharmaceuticals, and personal care industries. Perilla dried leaf extract is currently available on the market in various forms, including powder, tablets, and paste, and the demand for perilla leaf and its components is expected to grow significantly. However, there is limited research on fresh leaf extract, and data on its bioactive constituents and activities are lacking. Therefore, this study aims to compare the in vitro biological activities of fresh leaf extract (PLE_f_) and dry leaf extract (PLE_d_) of Thai perilla in terms of promoting anti-inflammatory activity in LPS-induced RAW 264.7 macrophage cells. The results of this study could provide valuable insights for the standardization and application of Thai perilla leaf extracts (PLEs) in the development of pharmaceuticals and nutraceuticals.

## 2. Results

### 2.1. Extraction Yields, TPC, TFC, and Phytochemical Contents in PLE_f_ and PLE_d_


The extraction yield, physical appearance, and phytochemical content of the Thai perilla leaf extracts (PLEs) are presented in Table 1. Although both extracts have similar physical appearances, PLE_d_ exhibited a higher extraction yield (7.9%) than PLE_f_ (3.9%). The TPC and TFC were higher in PLE_d_ compared to PLE_f_, with TPC at 748.0 ± 4.9 and 469.5 ± 4.2 mg GAE/g extract, and TFC at 977.0 ± 37.2 and 303.2 ± 11.8 mg CE/g extract for PLE_d_ and PLE_f_, respectively. 

Both PLEs contained hydrophilic phytochemical compounds, including rosmarinic acid, chlorogenic acid, caffeic acid, and ferulic acid, with rosmarinic acid being the predominant compound (Appendix A). PLE_d_ had high amounts of rosmarinic acid, chlorogenic acid, and caffeic acid, while PLE_f_ had relatively high amounts of ferulic acid and luteolin. 

### 2.2. Effect of PLEs on Scavenging of DPPH and ABTS Radicals

The antioxidant activities of PLEs were evaluated and are presented in Figure 1, indicating a dose-dependent scavenging of DPPH free radicals (Figure 1A). PLE_d_ exhibited higher antioxidant potential than PLE_f_, with IC_50_ values of 6.2 ± 0.3 μg/mL and 12.5 ± 1.3 μg/mL, respectively. Additionally, both extracts showed a dose-dependent suppression of ABTS^•+^ radicals (Figure 1B), with PLE_d_ exhibiting a higher suppression potential (IC_50_ = 1.1 ± 0.0 μg/mL) than PLE_f_ (IC_50_ = 2.1 ± 0.4 μg/mL). These findings are consistent with the higher TPC and TFC values of PLE_d_ compared to PLE_f_, indicating that the extracts may act as antioxidants to protect against oxidative stress-related conditions such as inflammation and carcinogenesis.

### 2.3. In Vitro Mutagenicity and Antimutagenicity Activity of PLE_f_ and PLE_d_

The genotoxic potential of PLEs was evaluated using an in vitro assay with *S. typhimurium* mutation. The results, shown in Table 2, indicate that PLE_f_ and PLE_d_ did not exhibit mutagenicity in either the TA98 or TA100 strains, with (+S9) or without (−S9) metabolic activation. Furthermore, all PLE_f_ and PLE_d_ concentrations tested in the experiment demonstrated no toxicity. 

In the presence of metabolic activation, the in vitro antimutagenic activity of PLEs against two food-borne carcinogens, PhIP and IQ, was evaluated, as shown in Table 3. PLE_f_ and PLE_d_ were found to have antimutagenic potential against both PhIP and IQ in Salmonella TA98 and TA100. PLE_f_ was more effective at reducing mutagenicity caused by PhIP and IQ than PLE_d_, and the inhibitory effect of the extracts on *S. typhimurium* mutation may be related to the enzymes involved in mutagen metabolism.

### 2.4. Cytotoxic Effect of PLEs on PBMCs and RAW 264.7 Cells

The cellular effects of PLEs were investigated by assessing their cytotoxicity on PBMCs and RAW 264.7 cells. After exposure to varying concentrations of PLE_f_ and PLE_d_ for 48 h, cell viability was found to remain unaffected at concentrations up to 100 μg/mL, with the percentage of cell viability in both PLE treatments exceeding 80% (Figure 2). PLE_f_ at a concentration of 200 μg/mL reduced PBMCs’ viability to 63% but did not affect RAW 264.7 cells (unreported data). The IC_20_ of PLE_f_ on PBMCs and RAW 264.7 cells were 152 ± 47 and >200 μg/mL, respectively. PLE_d_ exhibited lower cytotoxic effects on both PBMCs and RAW 264.7 cells than PLE_f_, with IC_20_ and IC_50_ values exceeding 200 μg/mL. Further experiments were conducted at PLE concentrations ranging from 0 to 100 μg/mL, as these concentrations did not significantly affect cell viability in any tested cell types.

### 2.5. Effect of PLEs on Reactive Oxygen Species (ROS) Generation in Human PBMCs 

PBMCs, isolated from healthy volunteers, were used as a model to examine the effect of PLEs on intracellular ROS generation. Intracellular ROS was detected using 2′,7′-dichlorofluorescein diacetate (DCFH-DA), which is oxidized to fluorescent dichlorofluorescein (DCF) by ROS [19]. Our findings indicate that PLEs have antioxidant potential and can inhibit intracellular ROS production in primary human PBMCs. The incubation of PBMCs with PLEs resulted in a significant dose-dependent reduction in ROS generation (Figure 3). Moreover, PLE_f_ exhibited a more efficient inhibition of ROS generation in PBMCs compared to PLE_d_, indicating its superior antioxidant potential. These results suggest that PLEs can act as antioxidants, mitigating the harmful effects of oxidative stress on cells and preventing intracellular damage.

### 2.6. Effect of PLEs on NO Production in LPS-Stimulated RAW 264.7 Cells

The anti-inflammatory activity of PLEs was assessed by measuring the production of NO in RAW 264.7 cells treated with LPS. The amount of NO released from the LPS-activated cells was quantified by measuring the accumulation of nitrite in the culture supernatant. PLEs were found to inhibit LPS-induced NO production in a dose-dependent manner without affecting cell viability in the RAW 264.7 cells (Figure 4). Specifically, at a concentration of 100 μg/mL, PLE_f_ significantly reduced NO production by 47%, while PLE_d_ showed a minor reduction in NO levels of 12%. The results indicate that PLEs exhibit anti-inflammatory properties, which likely make them useful for the treatment of inflammatory conditions.

### 2.7. Effect of PLEs on LPS-Induced iNOS and COX-2 Expression in RAW 264.7 Cells

The involvement of inducible nitric oxide synthase (iNOS) and cyclooxygenase-2 (COX-2) in cellular inflammation is well established. To evaluate the impact of PLEs on the mRNA and protein expression of iNOS and COX-2 in RAW 264.7 cells, reverse transcription quantitative polymerase chain reaction (RT-qPCR) and Western blot analysis were conducted. The results revealed that PLEs reduced the mRNA expressions of iNOS and COX-2 in a dose-dependent manner, with PLE_f_ being more effective than PLE_d_, as shown in Figure 5. Furthermore, Western blot analysis confirmed that PLEs dose-dependently decreased the protein levels of iNOS and COX-2, with PLE_f_ being similarly efficient to PLE_d_, as illustrated in Figure 6. Overall, these findings suggest that PLE_f_ and PLE_d_ have the potential to reduce LPS-induced inflammation by inhibiting the transcriptional expression of iNOS and COX-2, leading to a potential decrease in NO production in RAW 264.7 cells.

### 2.8. Effect of PLEs on LPS-Induced Pro-Inflammatory Cytokine Production and mRNA Expression in RAW 264.7 Cells

Pro-inflammatory cytokines, such as TNF-α and IL-6, are critical in inflammation cascades. This study investigated the impact of PLEs on pro-inflammatory cytokine production in LPS-induced RAW 264.7 cells using ELISA and RT-qPCR. The results revealed that PLEs potentially suppressed LPS-induced TNF-α and IL-6 production (Figure 7) by inhibiting their mRNA expressions (Figure 8A, B), as well as reducing IL-1β mRNA expression (Figure 8C) in LPS-treated RAW 264.7 cells. PLE_f_ showed a significant dose-dependent inhibition of TNF-α and IL-6 production and their mRNA expression, while PLE_d_ had slightly lower efficacy. Notably, IL-1β can enhance TNF-α and IL-6 mRNA expression and protein production. Thus, these findings suggest that PLEs may alleviate LPS-triggered inflammation in RAW 264.7 cells by suppressing TNF-α and IL-6 production through IL-1β reduction.

### 2.9. Effect of PLEs on LPS-Induced NF-κB Activation and c-Jun Production

To investigate the effects of PLEs on NF-κB and AP-1 activation induced by LPS, the phosphorylation of NF-κB and the nuclear translocation of AP-1 (c-Jun) were measured. The results showed that both PLE_f_ and PLE_d_ dose-dependently decreased NF-κB phosphorylation without affecting NF-κB production (Figure 9A). Specifically, PLE_f_ at 50 μg/mL and 100 μg/mL demonstrated a suppression of NF-κB phosphorylation by 15.5% and 28.7%, respectively. PLE_d_ at 100 μg/mL suppressed NF-κB phosphorylation by 5.6% compared to the control. However, there was no significant change in c-Jun nuclear translocation with PLEs, as shown in Figure 9B. These results demonstrate that PLE_f_ and PLE_d_ inhibit NF-κB activation by decreasing protein phosphorylation, leading to the downregulation of mRNA expression of IL-1β, TNF-α, IL-6, iNOS, and COX-2, as well as decreasing TNF-α, IL-6, and NO production.

## 3. Discussion

Thai perilla (Nga-mon) is a type of aromatic vegetable used in Thai cuisine that contains various phytochemicals such as rosmarinic acid, which is linked to its biological activities [20,21]. The present study demonstrated that perilla leaf extracts (PLEs) contain high amounts of phenolics and flavonoids. Specifically, PLE_d_ exhibited higher levels of TPC, TFC, and hydrophilic phytochemical compounds, including rosmarinic acid, chlorogenic acid, and caffeic acid, compared to PLE_f_. On the other hand, PLE_f_ contained a relatively high amount of ferulic acid and luteolin compared to PLE_d_. These differences in phytochemical content may be attributed to the form of the vegetal sample used in the extraction process, which can affect the presence of phytochemicals in the extracts.

Phytochemical studies commonly use fresh and dried herb samples, with dried samples preferred for their disinfectant, decontamination, and preservation properties [22]. However, the chemical and biological activities of herbs can be influenced by drying conditions [23,24], and enzymatic reactions in fresh plant cells can impact phytochemical metabolism, resulting in differences in phytochemical content. For example, drying was found to significantly affect the production of characteristic compounds in *R. fraxinifolius* leaves, with luteolin-7-O-glucuronide, an antioxidant flavonoid glycoside, being detectable in fresh extract but not in oven-dried extract [25]. In contrast, heat treatment was reported to enhance the antioxidant capacity of tamarind leaves by accelerating amine groups that scavenge singlet oxygen [26]. Comparison of fresh and dry leaf extracts from medicinal plants also revealed differences in chemical constituents such as phenolic compounds and flavonoid composition [21,26,27,28], consistent with our findings where phytochemical component in PLE_f_ and PLE_d_ were present in different ratios.

Kagawa et al., reported that the extract of fresh perilla leaves contained higher levels of rosmarinic acid, but lower amounts of luteolin and apigenin when compared to the extract of dried leaves [21]. In contrast, our study found that PLE_f_ had lower rosmarinic acid but higher luteolin levels than PLE_d_. However, our findings are consistent with those of Hossain and colleagues, who detected higher rosmarinic acid levels in the extract of dried leaves than in the extract of fresh leaves from six Lamiaceae herbs [29].

Reactive oxygen species (ROS) are produced in mammalian intracellular systems during the reduction of molecular oxygen, resulting in two significant endogenous sources of ROS: the mitochondrial electron transport chain and the cytochromes’ P450-dependent microsomal electron transport system. Uncontrolled ROS formation within cells can cause cellular or tissue damage and suppress the inflammatory response, which is linked to inflammatory and metabolic diseases [30]. Inhibiting intracellular ROS production and scavenging free radicals are potential strategies for reducing cellular oxidative damage and ameliorating pathogenesis. Natural polyphenols, such as those found in PLEs, which contain high levels of phenolics, flavonoids, and rosmarinic acid, have been shown to possess antioxidant properties and decrease cellular oxidative damage. In primary human PBMCs, PLEs have been found to inhibit intracellular ROS production and scavenge DPPH and ABTS free radicals in vitro. PLE_d_ exhibits higher antioxidant activity than PLE_f_, which correlates well with its higher TPC and TFC values. The findings suggest that PLEs may act as antioxidants, preventing cellular damage and providing a protective effect against oxidative stress, which is commonly associated with inflammation and carcinogenesis.

In addition to being a valuable source of natural antioxidants, PLEs have exhibited safety and antimutagenic activities that could contribute to their health benefits. When PBMCs and RAW 264.7 macrophage cells were exposed to PLE_f_ and PLE_d_ for 48 h, no cytotoxic effects were observed. Furthermore, an in vitro *S. typhimurium* reverse mutation assay showed that PLEs did not induce mutagenicity. Both PLE_f_ and PLE_d_ demonstrated antimutagenic potential against two food-borne carcinogens, PhIP and IQ. Human CYP1A2 selectively activates PhIP and IQ through N oxidation, which generates a critical metabolite that is implicated in genotoxicity and DNA adduct formation [31]. CYP enzymes are known to contribute to the production of intracellular ROS [32]. Elevated ROS levels can disrupt cellular redox homeostasis, leading to the oxidation of nucleic acids, DNA damage, and mutations, initiating carcinogenesis. Since ROS are formed during the metabolic processing of PhIP and IQ, the effective protection of PLEs against in vitro genotoxicity induced by PhIP and IQ may be attributed to their antioxidative activity.

Macrophages are vital in the body’s defense against infection and inflammation. Upon activation by lipopolysaccharides (LPSs), macrophages produce various immunostimulatory agents, such as interleukins (IL-1β, IL-6, IL-8), TNF-α, iNOS, NO, COX-2, and PGE_2_ [33]. Hence, anti-inflammatory agents often aim to target the inhibition of these pro-inflammatory mediators released by LPS-activated macrophages [34,35,36]. Pro-inflammatory cytokines such as IL-1β, IL-6, and TNF-α can activate NO production through inducible nuclear factors such as NF-κB and AP-1, leading to inflammation [37]. Overproduction of NO can activate COX-2, which is the rate-limiting enzyme in inflammation. Studies have shown that phenolic compounds, such as gallic acid, coumaric acid, and ferulic acid, can reduce pro-inflammatory cytokines and NO levels by inhibiting LPS-mediated NF-κB and iNOS expression in macrophages [38,39]. According to Lee and Han, the extract from dried leaves of Korean perilla can inhibit the expression of pro-inflammatory mediators such as IL-6, IL-1β, TNFα, iNOS, COX-2, and nuclear factor NF-κB in LPS-activated macrophages [40]. Our study also found that PLEs, especially PLE_f_, decreased NO production, inhibited iNOS, COX-2, TNF-α, and IL-6 mRNA and protein expression, and suppressed the mRNA expression of IL-1β in LPS-stimulated RAW 264.7 cells. Therefore, PLEs may effectively combat LPS-induced inflammation by reducing NO production in LPS-activated RAW 264.7 cells through downregulating iNOS and COX-2 expression at the transcriptional level, which is correlated with the suppression of TNF-α, IL-1β, and IL-6 gene and protein expressions.

NF-κB is a pleiotropic regulator of several genes that play a role in immune and inflammatory responses, including iNOS, COX-2, IL-1β, TNF-α, and IL-6. The expression of these genes is increased in LPS-stimulated RAW 264.7 cells. Recent studies have shown that phenolic acids, such as ferulic acid, p-coumaric acid, caffeic acid, and chlorogenic acid, can inhibit the phosphorylation of NF-κB and block the activation of the AP-1 transcription factor [33]. However, the molecular mechanisms associated with PLEs suppressing LPS-induced inflammation showed that treatment with PLEs restricted the phosphorylation and nuclear translocation of NF-κB p65 but did not decrease AP-1 (c-Jun) activation in LPS-stimulated RAW 264.7 cells. These results are consistent with those previously reported by Huang et al. [41], who demonstrated that pretreatment with perilla dried leaf extract restored the level of LPS-decreased cytosolic IκBα and inhibited the nuclear translocation of NF-κB. Overall, our results demonstrate that PLE_f_ and PLE_d_ inhibited NF-κB activation by reducing protein phosphorylation and translocation, which would downregulate the expression of IL-1β, TNF-α, IL-6, iNOS, COX-2, and NO. Consequently, PLEs have anti-inflammatory effects in LPS-induced RAW 264.7 cells.

The quantity of rosmarinic acid in perilla leaf extracts has been shown to correlate with the bioactivities of the extracts, as reported in previous studies [13,42]. However, our cell-based study revealed that PLE_f_, which has lower rosmarinic acid content but higher amounts of ferulic acid and luteolin, exhibited greater antimutagenic and anti-inflammatory activity than PLE_d_, the rosmarinic acid-rich extract. This can be partly attributed to the combination of hydrophilic bioactive compounds present in PLE_f_. Plant extracts contain various bioactive compounds, some of which may be unknown and co-exist with others, making it difficult to comprehend all the chemical and biological interactions that contribute to the final bioactivities. Therefore, it could be suggested that not only the content of rosmarinic acid in the extract but also the combination of other phenolic compounds may play important roles in determining the antimutagenicity and anti-inflammatory properties of PLEs. 

## 4. Materials and Methods

### 4.1. Chemicals 

Dimethylsulfoxide (DMSO), 2,2-diphenyl-1-picrylhydrazyl (DPPH), diammonium 2,2-azino-bis(3-ethylbenzothiazoline-6-sulfonate) (ABTS), 2′,7′-dichlorofluorescein diacetate (DCFH-DA), Griess reagent, 3-(4,5-dimethylthiazol-2yl)-2,5-diphenyltetrazolium bromide (MTT), lipopolysaccharide (LPS), rosmarinic acid, catechin, chlorogenic acid, ferulic acid, and caffeic acid were purchased from Sigma-Aldrich (St. Louis, MO, USA). Luteolin and apigenin were purchased from Chengdu Biopurify Phytochemicals Ltd. (Chengdu, Sichuan, China). Dulbecco’s modified Eagle’s medium (DMEM) and fetal bovine serum (FBS) were purchased from GIBCO-BRL, Invitrogen Co. (Grand Island, NY, USA). TNF-α and IL-1β Enzyme Link Immuno-Sorbent Assay (ELISA) kits were purchased from BioLegend (San Diego, CA, USA). Antibodies specific to COX-2, iNOS, phospho-NF-κB, NF-κB, c-Jun, and β-actin were purchased from Cell Signaling Technology Inc. (Beverly, MA, USA). Antibody specific to PARP was purchased from Santa Cruz Biotechnology (Santa Cruz, CA, USA).

### 4.2. Preparation of Thai Perilla Leaf Extracts

#### 4.2.1. Plant Materials

Thai perilla leaves were obtained from a local wholesaler in Nan, Thailand. Dr. Komsak Pintha and Dr. Payungsak Tantipaiboonwong collected and prepared a voucher specimen (code QSBG-K2) that has been verified by the Queen Sirikit Botanic Garden Herbarium in Chiang Mai, Thailand, for future use.

#### 4.2.2. Fresh Leaf Extraction 

Fresh Thai perilla leaves (100 g) were mixed with 1 L of 70% ethanol and stirred at room temperature for 12–18 h. The resulting mixture was filtered through filter paper, and the extract was then concentrated using a rotary evaporator at 40 °C and dried using a lyophilizer. This extract was named PLE_f_.

#### 4.2.3. Dry Leaf Extraction 

Thai perilla leaves were dried in a hot air oven at 60 °C for 12 h, then ground and sieved through a 0.05 mm mesh to obtain a uniform powder. The perilla dry leaf extract was prepared via the following method [43]. First, 100 g of the powder was mixed with 1 L of 70% ethanol and left to stir at room temperature for 12–18 h, resulting in an extract named PLE_d_. The extract was filtered, concentrated using a rotary evaporator at 40 °C, and dried using a lyophilizer. Both the PLE_d_ and PLE_f_ were stored at −20 °C for future use.

### 4.3. Total Phenolic and Total Flavonoid Content Determination

The Folin–Ciocalteu method was used to determine the total phenolic content (TPC) of PLEs, and the total flavonoid content (TFC) was determined using the aluminum chloride colorimetric method with slight modifications [43]. For TPC determination, the extracts were oxidized with Folin–Ciocalteu reagent and neutralized with 7% Na_2_CO_3_. After standing for 20 min in the dark, the absorbance at 760 nm was measured with a spectrophotometer. TPC was calculated using a standard curve obtained from various concentrations of gallic acid and expressed as mg of gallic acid equivalents (GAE)/g dry weight.

For TFC determination, the extract was mixed with 5% NaNO_2_ for 10 min, followed by the addition of 10% AlCl_3_∙6H_2_O and incubation for another 10 min. Afterward, 1 M NaOH was added, and the absorbance was measured at 532 nm. TFC was expressed as mg of catechin equivalents (CE)/g dry weight, using a standard curve based on different concentrations of catechin.

### 4.4. HPLC Analysis

The Agilent 1290 Infinity II was utilized to conduct HPLC analysis of PLEs, using a ZORBAX Eclipse Plus C18 column (5 μm, 4.6 × 150 mm) for gradient elution at 35 °C [43]. The mobile phase comprised 0.1% trifluoroacetic acid in water (A) and 100% methanol (B). Gradient elution was performed for 50 min from 100% to 0% A, followed by 5 min each of 100% B and 100% A to re-establish initial conditions before the next sample injection. The flow rate and injection volume were 1 mL/min and 10 μL, respectively. Monitoring was at 280 nm and 320 nm, with compound identification based on retention time and spectral matching. Quantification was achieved by comparing the peak areas of the samples with the calibration curves of corresponding standard solutions.

### 4.5. DPPH and ABTS Radical Scavenging

The antioxidant activity of PLEs was evaluated through DPPH and ABTS radicals scavenging assays, as described in our previous study [44]. For the DPPH assay, various concentrations of PLEs were mixed with a freshly prepared 0.2 mM DPPH radical solution and incubated with light protection for 20 min at room temperature. The decolorization of the DPPH radical was measured at 517 nm, and the antioxidant activity was expressed as % DPPH radical scavenging using Trolox as a standard reference.

For the ABTS assay, the various concentrations of PLEs were mixed with a diluted ABTS^•+^ solution (7 mM ABTS and 2.45 mM potassium persulfate at a 1:1 ratio, *v*/*v*) and incubated in the dark for 6 min. The absorbance of the reaction mixture was measured at 734 nm, and the antioxidant capacity was expressed as % ABTS radical scavenging using Trolox as a standard control.

### 4.6. Mutagenicity and Antimutagenicity Test

The mutagenic effects of PLEs were assessed using the S. typhimurium strains TA98 and TA100 with and without metabolic activation (+/−S9 mix) [44]. The tester strains were incubated at 30 °C with different concentrations of extracts and phosphate buffer or S9 mix. The top agar containing L-histidine and D-biotin was then added and poured onto a minimal glucose agar plate, and the number of histidine-independent revertant colonies was counted after 48 h of culture at 37 °C. Positive controls for the presence of metabolic activation (+S9) included 2-aminoanthracene (2-AA), amino-1-methyl-6-phenylimidazo[4,5-*b*]pyridine (PhIP), and 2-amino-3-methyl-3H-imidazo[4,5-*f*] quinolone (IQ), while 2-(2-furyl)-3-(5-nitro-2-furyl)-acrylamide (AF-2) was used for the absence of metabolic activation (−S9). The tests were performed in triplicate for each dose and repeated twice.

The antimutagenicity test was performed in the presence of metabolic activation, similar to the mutagenicity test. The Salmonella strains TA98 and TA100 were treated with a combination of PLEs and standard heterocyclic amine mutagens, 2-PhIP and IQ, respectively. The number of revertant colonies was counted after incubation and compared to treatment with the mutagen alone. The antimutagenicity was calculated and is expressed as a percentage of the inhibition of mutagenicity.

### 4.7. Cells and Cell Culture

The RAW 264.7 mouse macrophage cell line was obtained from the ATCC and cultured in Dulbecco’s modified Eagle’s medium (DMEM) supplemented with 10% heat-inactivated fetal bovine serum (FBS) and 1% penicillin/streptomycin at 37 °C in a 5% CO_2_ humidified atmosphere until reaching 80% confluence. 

Human peripheral blood mononuclear cells (PBMCs) were isolated using Ficoll-hypaque and then washed twice with ice-cold phosphate-buffered saline (PBS) at pH 7.4 before being resuspended in fresh RPMI medium.

### 4.8. Cell Viability Test

The assessment of cell viability was performed using the MTT colorimetric assay. Cells were exposed to various PLE concentrations in a 96-well plate and incubated at 37 °C for 48 h. Following incubation, MTT solution (5 mg/mL in PBS, pH 7.4) was added and the plates were incubated for another 4 h. The resulting formazan crystals were dissolved in DMSO after removing the medium, and absorbance was measured at 540 nm with a reference wavelength of 630 nm using an ELISA microplate reader.

### 4.9. Determination of ROS Production

To assess intracellular ROS production, PBMCs were treated with different concentrations of PLEs in a black 96-well plate. After a 24 h incubation at 37 °C, cells were washed twice with Hanks’ balanced salt solution (HBSS) and exposed to 20 µM DCFH-DA in HBSS for 30 min in the dark. Excess DCFH-DA was removed by washing twice with HBSS, and the cells were resuspended in 0.25 M NaOH. The fluorescence intensity of oxidized DCF was measured using a fluorescence spectrophotometer (excitation: 485 nm, emission: 530 nm) [45]. The experiments were conducted in triplicate, with 25 mg/mL EGCG serving as the positive control.

### 4.10. Determination of NO Production

To assess NO production, RAW 264.7 macrophage cells were exposed to different concentrations of PLEs for 2 h, followed by co-incubation with 1 µg/mL of LPS for 22 h. Following treatment, the culture medium was collected, and NO was measured using Griess reagent [45]. A standard curve generated through the serial dilution of NaNO_2_ in fresh culture medium was used to quantify the amount of nitrite present in the samples.

### 4.11. Measurement of Pro-Inflammatory Cytokines

To quantify TNF-α and IL-6 levels, sandwich ELISA assay kits from BioLegend (San Diego, CA, USA) were utilized. RAW 264.7 cells were initially seeded in a 6-well plate and incubated overnight. Following this, the cells were treated with various concentrations of PLEs for 2 h and then co-incubated with 1 µg/mL LPS for 22 h. The culture supernatants of the treated cells were collected and analyzed using ELISA kits for TNF-α and IL-6 according to the manufacturer’s instructions.

### 4.12. Total RNA Extraction and RT-qPCR

To measure the mRNA expression levels of TNF-α, IL-1β, IL-6, iNOS, and COX-2, RAW 264.7 cells were pretreated with varying PLE concentrations in a 6-well plate for 2 h. Subsequently, the cells were co-incubated with 1 µg/mL LPS for 22 h. The treated cells were collected, and total RNA was extracted using the NucleoSpin^®^ RNA kit. The RNA quantity was determined by measuring the absorption at 260 nm, and 1 µg RNA was reverse-transcribed into cDNA using the ReverTra Ace^®^ qPCR RT kit. The resulting cDNA was used for RT-qPCR amplification with the SensiFastTM SYBR^®^ Lo-ROX qPCR kit. The thermal cycling conditions consisted of an activation step at 95 °C for 10 min, followed by 40 cycles of 95 °C for 15 s and 60 °C for 60 s [45]. Target cDNA levels were normalized to GAPDH expression and are presented as relative expression levels compared to the LPS-treated control. The primers used in this study are listed in Table 4.

### 4.13. Preparation of Whole-Cell Lysate and Nuclear Fraction

To prepare the whole-cell lysate, RAW 264.7 cells were pretreated with different concentrations of PLEs for 12 h, followed by co-treatment with 1 µg/mL of LPS for 45 min. The treated cells were collected, washed twice with ice-cold PBS, and then incubated with RIPA buffer on ice for 20 min. The whole-cell lysate supernatant was collected via centrifugation at 12,000 rpm for 10 min [46]. 

To obtain the nuclear extract, treated cells were collected and washed twice with ice-cold PBS, then incubated with a hypotonic buffer on ice for 20 min. After adding 10% NP-40, the cell mixtures were vortexed for 15 s and centrifuged at 12,000 rpm for 5 min. The nuclear pellet was collected and suspended in ice-cold nuclear extraction buffer, then incubated on ice for 25 min. The nuclear mixture was then centrifuged at 12,000 rpm for 10 min, and the supernatant represented the nuclear fraction [46]. The protein concentration of the whole-cell lysate and nuclear extract was measured by utilizing the Bradford protein assay.

### 4.14. Western Blot Analysis

Both the whole-cell lysate and nuclear fraction were separated using SDS-PAGE and transferred onto a nitrocellulose membrane through electroblotting. The membrane was subsequently blocked with 5% skimmed milk in TBS containing 0.3% (*v*/*v*) Tween-20 for an hour and then incubated with specific primary antibodies at 4 °C overnight. After washing, the membrane was exposed to a secondary antibody for 2 h and then detected using chemiluminescence.

### 4.15. Statistical Analysis

Statistical analysis was performed using GraphPad Prism 8.0 software, with ANOVA being used to compare multiple groups and the Tukey multiple-comparison test used to determine significant differences. Statistical significance was indicated by * *p* < 0.05, ** *p* < 0.01, and *** *p* < 0.001.

## 5. Conclusions

In this study, Thai perilla extracts demonstrate potent antioxidant, antimutagenic, and anti-inflammatory properties by scavenging free radicals, inhibiting intracellular ROS production, and suppressing NF-κB p65 phosphorylation and nuclear translocation, which leads to the downregulation of proinflammatory mediators such as IL-1β, TNF-α, IL-6, iNOS, COX-2, and NO. PLE_f_, with lower rosmarinic acid but higher ferulic acid and luteolin, shows higher antimutagenic and anti-inflammatory activity than PLE_d_, indicating that the mechanisms of PLEs are not solely dependent on rosmarinic acid content, but rather on the combination of natural bioactive phytoconstituents. PLEs offer promising potential as natural and safe extracts for antioxidant, anti-inflammatory, and chemopreventive applications, highlighting a possible approach to promote human health or incorporate them as dietary supplements.

## Figures and Tables

**Figure 1 plants-12-02210-f001:**
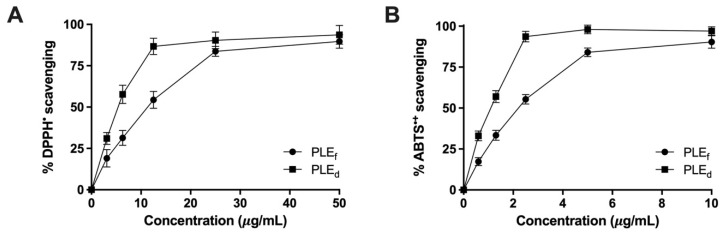
Antioxidant capacity of PLEs: (**A**) the scavenging of DPPH and (**B**) ABTS radicals. Data represent the mean of three independent triplicate experiments (*n* = 3). Error bars indicate SD.

**Figure 2 plants-12-02210-f002:**
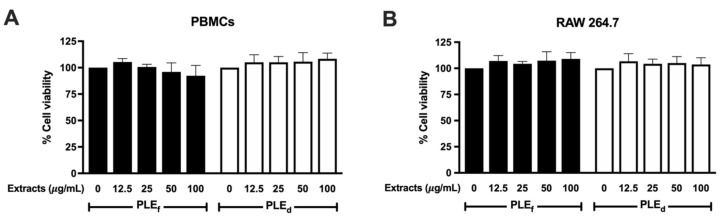
The cytotoxic effect of PLE_f_ and PLE_d_ on (**A**) PBMCs and (**B**) RAW 264.7 cells at 48 h. Data represent the mean of three independent triplicate experiments (*n* = 3). Error bars indicate SD.

**Figure 3 plants-12-02210-f003:**
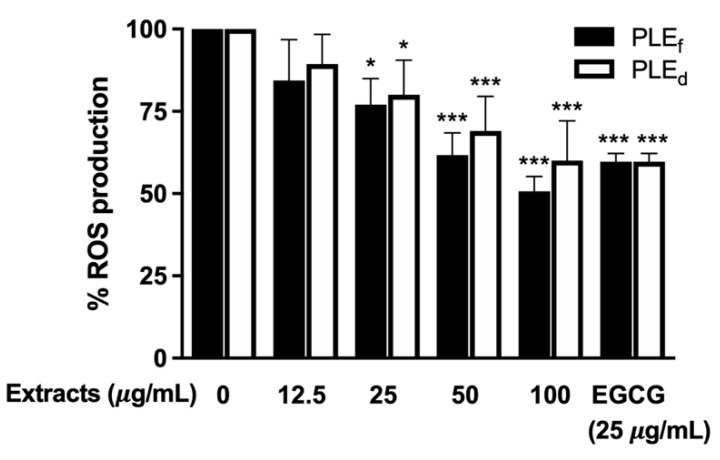
Intracellular ROS production in PBMCs. Data represent the mean of three independent triplicate experiments (*n* = 3). Error bars indicate SD. * *p* < 0.05, *** *p* < 0.001 versus no extract treatment.

**Figure 4 plants-12-02210-f004:**
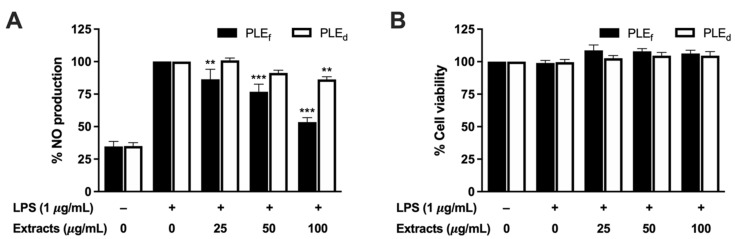
Effect of PLEs on (**A**) NO production and (**B**) cell viability of LPS-treated RAW 264.7 cells. Data represent the mean of three independent triplicate experiments (*n* = 3). Error bars indicate SD. ** *p* < 0.01, *** *p* < 0.001 versus LPS without extract treatment.

**Figure 5 plants-12-02210-f005:**
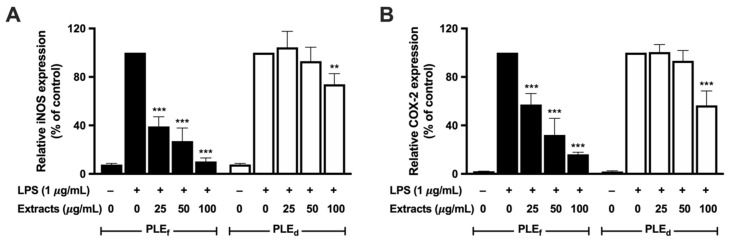
Effect of PLEs on LPS-induced mRNA expression of (**A**) iNOS and (**B**) COX-2 in RAW 264.7 cells. Data represent the mean of three independent triplicate experiments (*n* = 3). Error bars indicate SD. ** *p* < 0.01, *** *p* < 0.001 versus LPS without extract treatment.

**Figure 6 plants-12-02210-f006:**
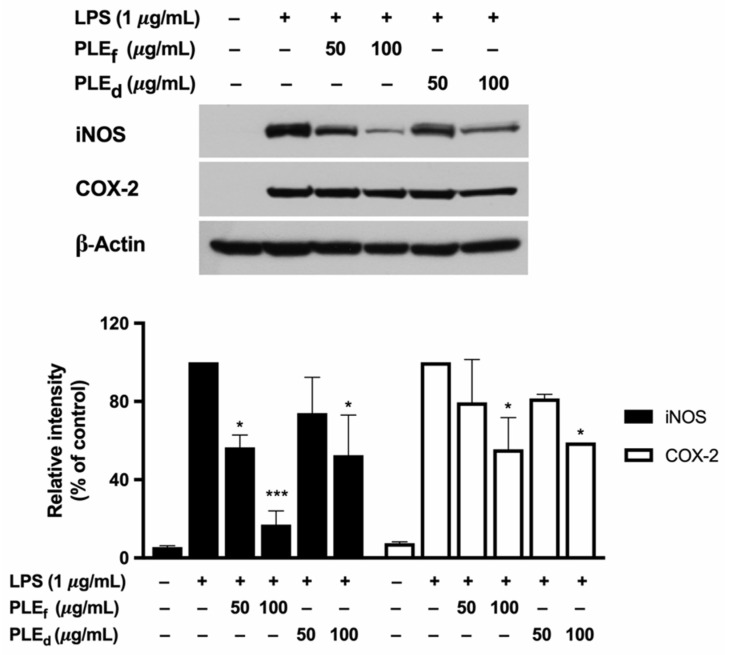
Effect of PLEs on LPS-induced iNOS and COX-2 production in RAW 264.7 cells. The cells were pretreated with different concentrations of PLEs for 2 h and then co-treated with 1 µg/mL of LPS for 22 h. iNOS and COX-2 levels in whole-cell lysate were detected through the Western blot analysis, and the data from a typical experiment are presented. Similar results were obtained from two independent experiments (*n* = 2). Error bars indicate SD. * *p* < 0.05, *** *p* < 0.001 versus LPS without extract treatment.

**Figure 7 plants-12-02210-f007:**
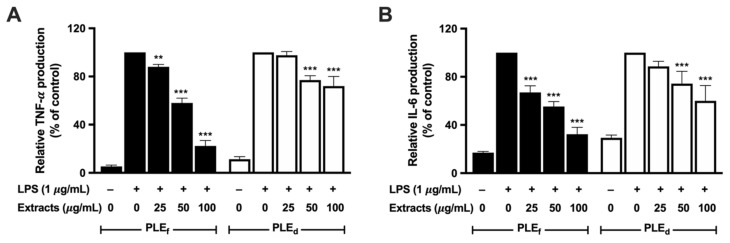
Effect of PLEs on LPS-induced (**A**) TNF-α and (**B**) IL-6 production in RAW 264.7 cells. Data represent the mean of three independent triplicate experiments (*n* = 3). Error bars indicate SD. ** *p* < 0.01, *** *p* < 0.001 versus LPS without extract treatment.

**Figure 8 plants-12-02210-f008:**
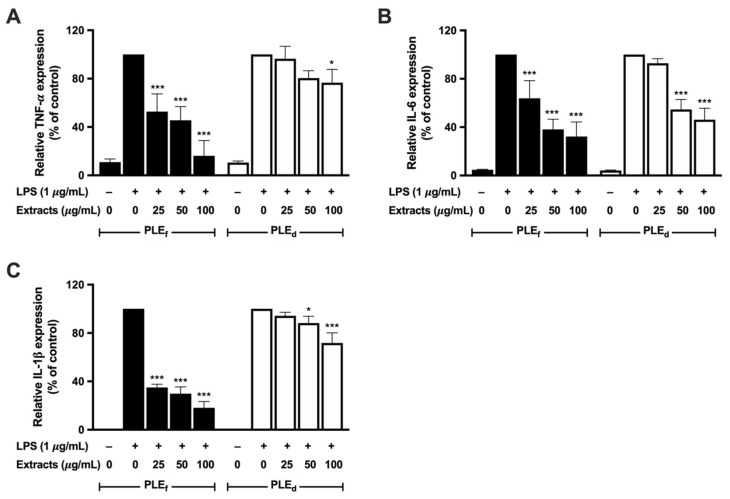
Effect of PLEs on LPS-induced (**A**) TNF-α, (**B**) IL-6, and (**C**) IL-1β mRNA expression in RAW 264.7 cells. Data represent the mean of three independent triplicate experiments (*n* = 3). Error bars indicate SD. * *p* < 0.05, *** *p* < 0.001 versus LPS without extract treatment.

**Figure 9 plants-12-02210-f009:**
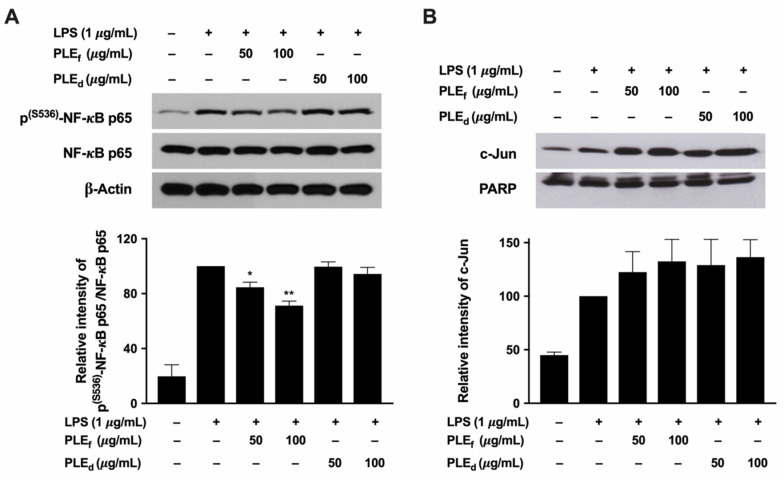
Effects of PLEs on LPS-induced (**A**) NF-κB and (**B**) AP-1 activation in RAW 264.7 cells. The cells were pretreated with various concentrations of PLEs for 12 h and then co-treated with 1 μg/mL of LPS for 45 min. The total NF-κB p65 and its phosphorylation levels in the whole-cell lysate were detected through Western blot analysis, and the data from a typical experiment are presented, while similar results were obtained from three independent experiments (*n* = 3). Nuclear extracts were prepared in order to analyze the nuclear translocation of AP-1 (c-Jun), and the data represent the mean of two independent experiments (*n* = 2). Error bars indicate SD. * *p* < 0.05, ** *p* < 0.01 versus LPS without extract treatment.

**Table 1 plants-12-02210-t001:** Extraction yield and phytochemical screening of PLE_f_ and PLE_d_.

Contents	PLE_f_	PLE_d_
Extract yield (%*w*/*w*)	7.4	14.2
Physical appearance	Dark green-brownish powder	Green-brownish powder
TPC (mg GAE/g extract)	469.5 ± 4.2	748.0 ± 4.9
TFC (mg CE/g extract)	303.2 ± 11.8	977.0 ± 37.2
Phytochemical contents * (mg/g extract)		
- Rosmarinic acid	1.38 ± 0.01	23.57 ± 0.30
- Chlorogenic acid	0.56 ± 0.08	0.95 ± 0.01
- Caffeic acid	0.21 ± 0.00	0.50 ± 0.01
- Ferulic acid	0.50 ± 0.01	0.38 ± 0.04
- Luteolin	0.16 ± 0.01	ND

* The values are expressed as means ± SD from duplicated results (*n* = 2). ND = Not detected.

**Table 2 plants-12-02210-t002:** In vitro mutagenicity of PLEs in *S. typhimurium* strains TA98 and TA100.

Treatment	Concentration (µg/Plate)	Number of Revertants/Plate (MI)			
		TA98		TA100	
		−S9	+S9	−S9	+S9
Vehicle control:					
DMSO		26 ± 4	35 ± 5	108 ± 4	117 ± 3
Positive control:					
2-AA	0.25	–	415 ± 22 (11.86)	–	605 ± 21 (5.17)
PhIP	1.00	–	530 ± 29 (15.14)	–	–
IQ	0.50	–	–	–	793 ± 28 (6.78)
AF-2	0.01	–	–	513 ± 64 (4.75)	–
	0.10	341 ± 18 (13.12)	–	–	–
PLE_f_	50	21 ± 5 (0.81)	30 ± 6 (0.86)	106 ± 6 (0.98)	111 ± 8 (0.95)
	100	24 ± 4 (0.92)	33 ± 4 (0.94)	111 ± 10 (1.03)	115 ± 9 (0.98)
	200	25 ± 3 (0.96)	29 ± 5 (0.83)	103 ± 6 (0.95)	112 ± 9 (0.96)
	400	23 ± 6 (0.88)	35 ± 7 (1.00)	110 ± 4 (1.02)	115 ± 4 (0.98)
PLE_d_	50	23 ± 4 (0.88)	35 ± 9 (1.00)	109 ± 4 (1.01)	118 ± 7 (1.01)
	100	26 ± 5 (1.00)	34 ± 9 (0.97)	110 ± 10 (1.02)	114 ± 8 (0.97)
	200	25 ± 4 (0.96)	29 ± 2 (0.83)	112 ± 7 (1.04)	115 ± 5 (0.98)
	400	24 ± 3 (0.92)	32 ± 3 (0.91)	112 ± 9 (1.04)	117 ± 9 (1.00)

MI, mutagenic index = number of revertant colonies on test plate/number of spontaneous revertant colonies. The results are expressed as means ± SD of two independent triplicate experiments (*n* = 2).

**Table 3 plants-12-02210-t003:** In vitro antimutagenicity of PLEs in *S. typhimurium* strains TA98 and TA100.

Treatment	Concentration (µg/Plate)	TA98		TA100	
		Number of Revertants/Plate	% Inhibition of Mutagenesis	Number of Revertants/Plate	% Inhibition of Mutagenesis
Standard mutagen:					
PhIP	1.0	530 ± 29	–	–	–
IQ	0.5	–	–	793 ± 28	–
Std. mutagen + PLE_f_	50	250 ± 22	53	523 ± 20	34
	100	159 ± 10	70	340 ± 21	57
	200	83 ± 10	84	232 ± 16	71
	400	58 ± 9	89	186 ± 20	76
Std. mutagen + PLE_d_	50	363 ± 17	32	616 ± 10	22
	100	273 ± 39	49	491 ± 15	38
	200	166 ± 19	69	339 ± 13	57
	400	88 ± 11	83	239 ± 12	70

The results are expressed as means ± SD of two independent triplicate experiments (*n* = 2).

**Table 4 plants-12-02210-t004:** Primers used for the RT-qPCR.

Gene	Sequence (5′ to 3′)
TNF-α	Fw: CGGGCAGGTCTACTTTGGAG
	Rv: ACCCTGAGCCATAATCCCCT
IL-1β	Fw: AAAAAAGCCTCGTGCTGTCG
	Rv: GTCGTTGCTTGGTTCTCCTTG
IL-6	Fw: GTTCTCTGGGAAATCGTGGA
	Rv: TGTACTCCAGGTAGCTATGG
iNOS	Fw: GCCACCAACAATGGCAACAT
	Rv: TCGATGCACAACTGGGTGAA
COX-2	Fw: TGAGCACAGGATTTGACCAG
	Rv: CCTTGAAGTGGGTCAGGATG
GAPDH	Fw: CACTCACGGCAAATTCAACGGC
	Rv: GACTCCACGACATACTCAGCAC

## Data Availability

Data are contained within the article.

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
