# Peer review of "Bioefficacy of Nga-Mon (Perilla frutescens) Fresh and Dry Leaf: Assessment of Antioxidant, Antimutagenicity, and Anti-Inflammatory Potential"

_plants, 2023, doi:10.3390/plants12112210_

Round 1

Reviewer 1 Report

The manuscript "Bioefficacy of Nga-Mon (Perilla frutescens) Fresh and Dry Leaf: Assessment of Antioxidant, Antimutagenicity, and Anti-inflammatory Potential" by P. Tantipaiboonwong et al. is devoted to the phytochemical investigation of Perilla frutescens leaf extracts. The authors described the extraction, chemical analysis and biochemical assays (radical scavenging, intracellular ROS generation, antimutagenic activity and LPS-induced inflammation) of the fresh and dried leafs' extracts. The work is written well, easy to read and could be accepted for publication after minor revision.

1.  I guess it'll be better for perception if the number of abbreviations used in the abstract will be reduced. For example:

in DPPH and ABTS radical scavenging - in radical scavenging tests (assays);

activity against PhIP and IQ in S. typhimurium TA98 and TA100 - activity in S. typhimurium model strains.

2. The presence of volatile compounds could be important factor especially for the material used in food industry. This parameter should differs for fresh and dried leaf extracts. Could the authors provide any measurements clarifying this issue? I guess it should be discussed in the manuscript.

Reviewer 2 Report

In the materials & methods section:

1-the authors didn't cite any reference for the assays 

2- The reference standards used in HPLC, TPC & TFC (the source is not mentioned= whether purchased or isolated

3- In the method of extraction of plants: the authors didn't mention why using 70% ethanol, evaporation via rotary vapor the temperature is not mentioned

4- The drying leaves in an oven at 60 oC could affect the secondary metabolites, you'd better used a vacuum oven to make a fair comparison

Results section

-The TFC in the dried leaves is higher than that of the TPC, however, flavonoids are calculated in the TP as a part of the phenolic content. Needs explanation

- The HPLC chromatograms are not displayed to show the difference between the two extracts. Should be displayed against  the retention time of the available standards 
